# Modeling of Groundwater Nitrate Contamination Due to Agricultural Activities—A Systematic Review

Meenakshi Rawat , Rintu Sen , Ikenna Onyekwelu, Travis Wiederstein and Vaishali Sharda *

Department of Biological and Agricultural Engineering, Kansas State University, Manhattan, NY 66502, USA
* Correspondence: vsharda@ksu.edu; Tel.: +1-(785)-532-2745

**Abstract:** Groundwater nitrate contamination is a significant concern in agricultural watersheds worldwide with it becoming a more pervasive problem in the last three decades. Models are great tools that are used to identify the sources and spatial patterns of nitrate contamination of groundwater due to agricultural activities. This Systematic Review (SR) seeks to provide a comprehensive overview of different models used to estimate nitrate contamination of groundwater due to agricultural activities. We described different types of models available in the field of modeling groundwater nitrate contamination, the models used, the input requirements of different models, and the evaluation metrics used. Out of all the models reviewed, stand-alone process-based models are predominantly used for modeling nitrate contamination, followed by integrated models, with HYDRUS and LEACHM models being the two most commonly used process-based models worldwide. Most models are evaluated using the statistical metric Root Mean Square Error (RMSE) followed by the correlation coefficient (r). This study provides the current basis for model selection in modeling nitrate contamination of groundwater due to agricultural activities. In addition, it also provides a clear and concise picture of the state of the art and implications to the scientific community doing groundwater quality modeling studies.

**Keywords:** agriculture; groundwater; HYDRUS; model evaluation metrics; nitrate; process-based models; systematic review

## 1. Introduction

Due to the increasing global population, food demand is expected to increase between 59% to 98% by 2050 [1]. There is a need to boost food production for the growing population and in addition to irrigation, modern agriculture depends majorly on fertilizer as a low-cost and low-effort input to achieve this [2–4]. Nitrogen is an essential nutrient for crop growth and development. Therefore, adequate supply of nitrogen fertilizers in the form of urea and NPK (Nitrogen Phosphorus Potassium) plays a crucial role in crop production [5,6].

In recent years, the overconsumption of nitrogen fertilizers has resulted in groundwater contamination which has become a major environmental concern [7,8]. Groundwater is a source of drinking and irrigation for 50% and 43% of the world's population, respectively [9]. Past studies have found that agriculture contributes to 60% of groundwater contamination globally [10], and that the amount of nitrogen fertilizer applied is strongly correlated to high nitrate concentrations in groundwater [11]. The consumption of nitrate-contaminated groundwater leads to several health issues [12–14], and studies have found an increase in the risk of diseases like methemoglobinemia or blue baby syndrome in newborn babies due to consumption of nitrate-contaminated water [15,16].

Concerns about nitrate-contaminated groundwater from agricultural activities have been the topic of several research studies that have included onsite field monitoring, intensive lab testing, and computer simulations [17–20]. Some existing studies have reported generalized approaches of nitrate contamination estimation and its impact on groundwater. For example, a study [21] showed that different management strategies impacted nitrate

leaching in orchard fields. Another study [22] showed the impact of climate change on future nitrate concentrations of groundwater of the UK. Similarly, [23] discussed the problems of nitrogen input into groundwater. Field experiments and lab tests are labor-intensive, tedious, and expensive, but provide the data required for model parameterization for the models to simulate the field environment. These studies have covered water quantity control, socioeconomic issues, management strategies, and climate change impacts, but have not discussed about model usage for groundwater nitrate contamination due to agricultural activities.

The use of computer models as a way to quantify nitrate contamination of groundwater due to agricultural activities has also been reported in some studies around the world. Out of these, few studies focus on how to reduce nitrate contamination from agricultural activities whereas others include scenario analyses or different management practices [24–26]. As the adoption of new crop management practices gains popularity, computer models prove to be useful tools to simulate field conditions and test the impacts of management practices on curbing nitrate contamination of groundwater. A study reported the [27] use of the computer simulation model, HYDRUS-2D, to estimate the amount of nitrate leaching from an agricultural field and found that a well-calibrated and validated model can prove to be a useful tool to provide a precise estimation of contamination. In another study, De Nitrification-De Composition (DNDC) model was calibrated for an agriculturally intense region in Northern China and it was found that the model is capable to capture the explicit spatial patterns of nitrate leaching and nitrogen fluxes in the study area due to spatially variable fertilizer application rates [28]. Likewise, according to [29], in the Hubei Province of China, nitrate concentration in the subsurface region increased with increasing nitrogen application which has been measured using calibrated and validated HYDRUS-2D model for lateral saturated flow and vertical leaching of nitrate. SWAT model was used to simulate crop yield and nitrate contamination for corn-peanut crop rotation under a variety of irrigation and nutrient management practices and found that sensor-based irrigation results in 40% less nitrate contamination under a 45% irrigation application without significant change in yield [30]. The GLEAMS model was used to quantify the impact of irrigation on nitrate contamination under potato farming [31]. The study concluded that additional irrigation water for frost prevention enhanced groundwater nitrate contamination. Similarly, [32] studied transport and fate of nitrate within the soil profile and nitrate leaching to subsurface drains in intensive agriculture farmland by comparing field data with simulation results obtained from LEACHMN model. They found that LEACHMN model performed satisfactorily in simulating nitrate in soil and subsurface drainange at the field scale. In addition, they concluded that LEACHMN model after calibration is a useful tool to demonstrate the distribution and transport of nitrate in groundwater due to agricultural activities. All these studies showed that the use of computer models were useful and efficient in estimating groundwater contamination due to agricultural activities.

Though computer models are an effective way to assess nitrate contamination of groundwater, there are several options available when it comes to choosing the model that best suits the needs of a specific research area. Given the existence of a range of models that can be used to simulate groundwater nitrate contamination, there is a need to determine which models are commonly used in the domain of groundwater nitrate contamination due to agricultural activities and what input variables are required by different models to simulate the nitrate contamination levels.

To address this issue, we conducted a Systematic Review (SR) to provide succinct updates on the state of groundwater nitrate contamination modeling due to agricultural activities. Therefore, this SR study aims to provide detailed information on different kinds of models used in modeling groundwater nitrate contamination due to agricultural activities. This SR aims to gain a detailed insight into the topic based on publications available worldwide. For this purpose, the following objectives/research questions were defined:

i.   What models have been used in literature to estimate nitrate contamination of groundwater?

ii.     Which input variables are needed by the models to simulate groundwater nitrate contamination?

iii.    What models have been commonly used worldwide in this area?

iv.     What statistical/evaluation metrics have been used to evaluate the performance of models?

v.      What are the challenges faced in using models for estimating nitrate contamination of groundwater due to agricultural activities?

## 2. Methods

This SR was designed to provide a clear insight on the models used around the world for simulating groundwater nitrate contamination due to agricultural activities. For this purpose, Preferred Reporting Items for Systematic Reviews and Meta-Analysis (PRISMA) guidelines [33] were followed and a SR methodology was defined which includes identification, screening, eligibility criteria, and full-text assessment. The details of each step of the SR methodology are as follows:

### 2.1. Identification

Groundwater nitrate contamination is a broad term and there could be several different types of contaminants (fertilizers, pesticides, heavy metals, etc.) that contribute towards it along with different sources of contamination (agricultural as well as industrial). For this study, we focus specifically on nitrate contamination of groundwater due to agricultural activities. To provide knowledge and background of different types of models used for the estimation of groundwater nitrate contamination, keywords were identified to get the relevant publications. Accurate keyword selection is key to any good SR study, and to achieve this selected keywords were converted into a string. The string was defined by narrowing it down to the basic concepts that are relevant to the scope of this study. The following string was used with AND and OR connection:

"Groundwater" AND "Nitrate" AND "Agriculture" AND "Modeling" AND ("Contamination" OR "Pollution" OR "Leaching" OR "Blue Baby Syndrome")

The literature search was performed for relevant studies published between 1 January 2000, and 31 December 2022. Only studies that were written in English languages were included because English is the most used language worldwide. After identifying the string, publication period, and specified language, the most used electronic databases were selected. The electronic databases used for this SR study are as follows: Google Scholar, World Cat, Science Direct, Scopus, Web of Science, JSTOR, EBSCO, ProQuest, and Taylor & Francis. The string defined above was then searched in the selected electronic databases along with language and publication period constraints. Figure 1 represents the number of publications obtained from the selected electronic databases in ascending order.

As shown in Figure 1, most of the publications were obtained from Science Direct and Google Scholar. This was expected since these are the largest literature sources and full-text electronic databases that provide scientific works from all over the world in diverse scientific fields.

### 2.2. Screening

After identifying all relevant publications in the previous step, the publications were screened using two criteria as defined below:

Criterion I—Duplicates removal: All selected titles were added to an MS Excel [34] sheet and duplicate titles were removed.

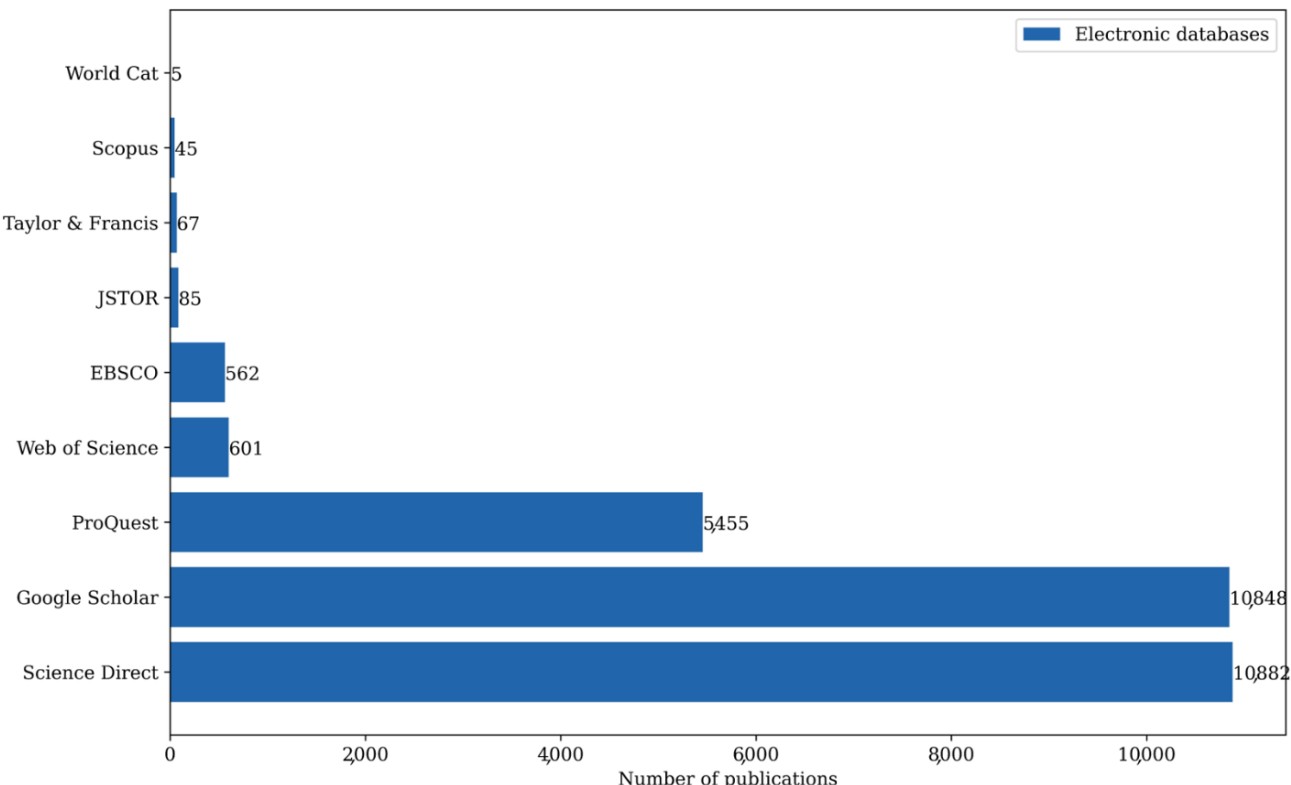

**Figure 1.** Number of publications "identified" from selected electronic databases.

Criterion II—Peer-reviewed publications: After removing the duplicates, peer-reviewed publications were selected. Given the broad nature of the domain of this search, finding all relevant publications by manually searching through conferences and journals would be very time-consuming. We, therefore, opted to start the search process with an automated search. We limited the search to electronic databases only and solely considered peer-reviewed journals. This created a structured search process that required rigorous methods to ensure that the results are both reliable and meaningful.

After the screening step, the selected publications were checked against the eligibility criteria developed in the next step.

*2.3. Eligibility Criteria*

In this step, the publications were analyzed based on inclusion and exclusion criteria to set the boundaries for the SR. The publications were filtered for the title, abstract, and then full text. For this purpose, two exclusion criteria were defined which are as follows:

Exclusion Criterion I (EC I):—For titles and abstracts filtering, we read the titles and abstracts to get the relevant publications that covered groundwater nitrate contamination and the use of modeling techniques to estimate contamination, anywhere in the world. This lead to a subset of publications for the full text reading. When we started reading the full text, we encountered few publications that reported research on the modeling of nitrate contamination of groundwater but specifically did not mention agricultural activities as the main source of contamination. We also found few publications that conducted a comparison of their approach to modeling groundwater contamination with other studies. This could be possibly caused due to the usage of large string in the databases. We found that the publications we obtained had either OR or AND terms but rarely both and that the search string lacked OR and AND keywords explicitly. To solve this problem, we restricted the search keywords after reading few full-text publications. To get the publications that satisfy these criteria, and to make our methodology more robust, the string was modified to include all the AND keywords and described as exclusion criteria II.

Exclusion Criterion II (EC II): The following string was finalized using four defined keywords and connected with the AND term:

"Agriculture" AND "Nitrate" AND "Modeling" AND "Groundwater"

The string was then searched in all the publications obtained after EC I. Publications that had all four keywords of the string were kept for further analysis, and the remaining publications were discarded.

### 2.4. Full-Text Assessment

The publications that satisfied EC II were selected for full-text assessment. These publications were used to show broader trends of model types, input variables required by models, evaluation metrics used by models, and the most commonly used model. Based on this SR methodology (see Figure 2), 28,550 publications were identified in the first step out of which, 21,700 publications were shortlisted for the title and abstract filtering after removing duplicates and non-peer-reviewed articles. Title and abstract filtering were done by using EC I in the third step, bringing down the number of publications to 233. After applying EC II, we obtained 75 publications for full-text assessment but found that out of these, 19 publications were not accessible. So, we were left with 56 publications in the end. These 56 publications were further analyzed to answer the research questions of this study.

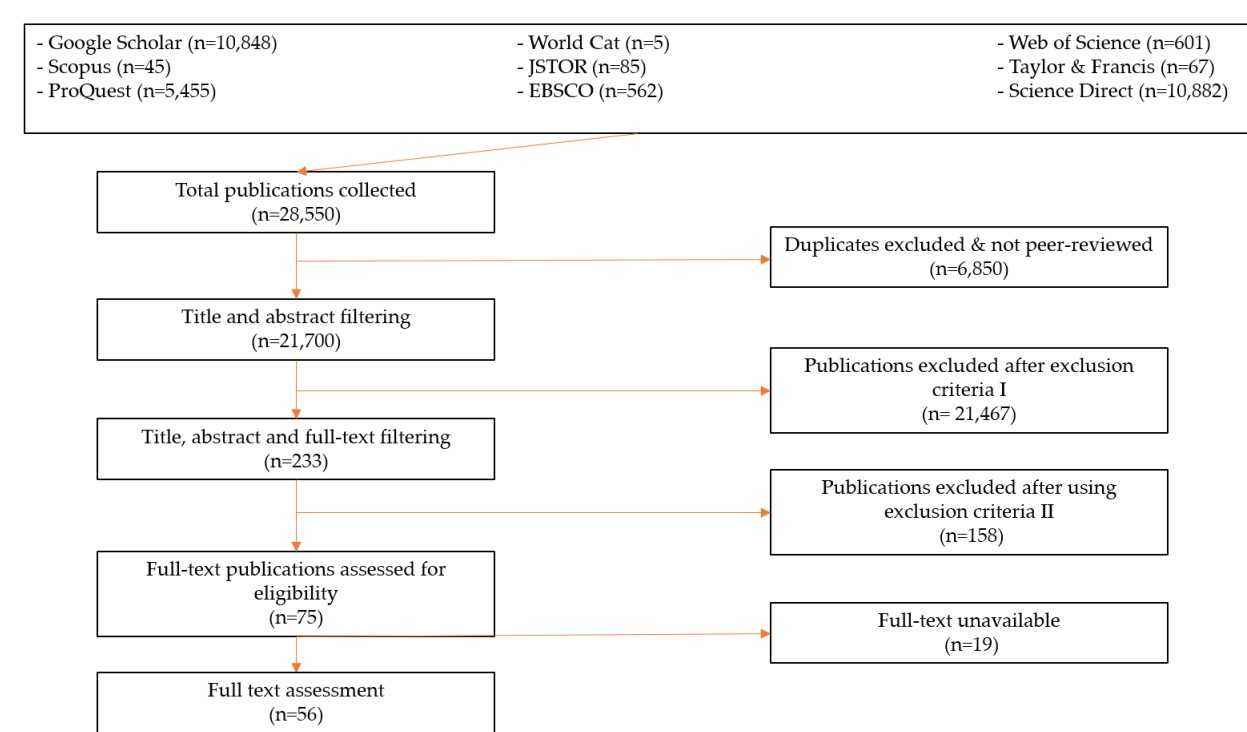

**Figure 2.** A detailed flowchart showing the methodology steps used for this SR adapted from Preferred Reporting Items for Systematic Reviews and Meta-Analysis (PRISMA) with number of publications in parenthesis.

## 3. Results

### 3.1. Literature Selection and Distribution

Figure 3 shows the distribution of studies collected from all over the world following the criteria defined above. The vast majority of publications reporting modeling of nitrate contamination of groundwater due to agricultural activities have been conducted in China followed by the United States. We observe that 23.7% of the publications are based in China, 16.9% in the United States, 6.7% in Iran, and Spain, 5% in Canada, and, Korea, Japan, Italy, Greece, Germany, France, and Egypt, together accounted for 25.4%, with the rest of the publications (16.9%) distributed among other countries of the world. These

numbers point towards the higher use of modeling as a way to understand the issues of nitrate contamination of groundwater in various parts of the world. According to [35], studies on yield optimization and environmental conservation are more common in the US, China, European Union, and Canada, which could explain why modeling of nitrate contamination of groundwater is concentrated in these regions. They also argued that these countries of the world may have stood out due to better allocation of resources towards research and development of agricultural productivity. Several studies around the world focused on nitrate contamination of groundwater, ranging from piezometric groundwater sampling, isotope tracing of nitrate leaching potentials, chemical analyses, and water quality assessments [36–40]. However, they were not included in this SR since we focused on the use of modeling to study nitrate contamination of groundwater due to agricultural activities.

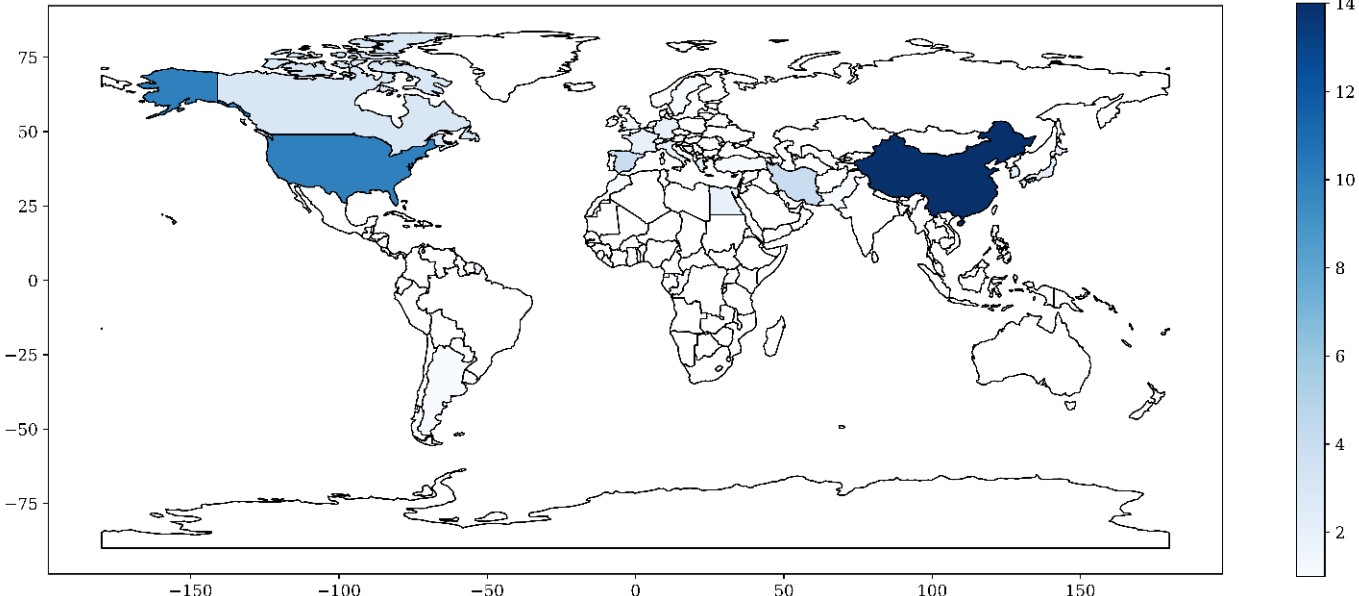

**Figure 3.** Spatial distribution of publications on the topic of modeling groundwater nitrate contamination due to agricultural activities across the world.

Table 1 shows the publication year, title and the model used. Figure 4 shows yearly publication distributions from 2000 through 2022, with 2013 being the year with the highest number of publications, followed by 2011 and 2021, respectively, and no publications on the selected topic in 2018. Although the temporal distribution of publications selected for this study (Figure 4) did not follow a specific trend or distribution, it is important to note that the total number of publications on the topic of this SR was higher in the recent decade (2011–2022) as compared to the decade before it (Figure 4). This is an indication that the use of computer models to simulate groundwater nitrate contamination due to agricultural activities is gaining research attention.

**Table 1.** Publications selected for this SR with the year of publication, title, type of model used, and the reference.

| Year | Title | Model Used | References |
|------|-------|------------|------------|
| 2000 | Modeling and testing of the effect of tillage, cropping and water management practices on nitrate leaching in clay loam soil | LEACHM and statistical | [41] |
| 2001 | Regional nitrate leaching variability: what makes a difference in northeastern Colorado | NLEAP | [42] |

**Table 1.** *Cont.*

| Year | Title | Model Used | References |
|------|-------|-----------|-----------|
| 2001 | Modeling the effect of chemical fertilizers on ground water quality in the Nile Valley aquifer, Egypt | GWS-3D | [43] |
| 2002 | Linkage of a geographical information system with the gleams model to assess nitrate leaching in agricultural areas | GLEAMS + PC-Arc/Cad GIS | [44] |
| 2003 | Modeling nitrogen dynamics in unsaturated soils for evaluating nitrate contamination of the Mnasra groundwater | Mathematical model | [45] |
| 2003 | Simulation of nitrate leaching for different nitrogen fertilization rates in a region of Valencia (Spain) using a GIS-GLEAMS system | GIS-GLEAMS | [24] |
| 2004 | Assessment of groundwater contamination by nitrate leaching from intensive vegetable cultivation using geographical information system | GIS | [46] |
| 2005 | A technique to estimate nitrate nitrogen loss by runoff and leaching for agricultural land, Lancaster County, Nebraska | NRCS-CN model | [47] |
| 2005 | Simulation of nitrogen leaching in sandy soils in The Netherlands with the ANIMO model and the integrated modelling system STONE | ANIMO, and STONE | [48] |
| 2005 | Modeling nitrogen uptake and potential nitrate leaching under different irrigation programs in nitrogen-fertilized tomato using the computer program NLEAP | NLEAP | [18] |
| 2005 | Nitrate leaching in cottonwood and loblolly pine biomass plantations along a nitrogen fertilization gradient | LEACHMN | [49] |
| 2006 | Evaluation of urea-ammonium-nitrate fertigation with drip irrigation using numerical modeling | HYDRUS-2D | [50] |
| 2006 | Nitrogen fertilization and nitrate leaching into groundwater on arable sandy soils | Numerical model | [51] |
| 2007 | Modeling nitrate contamination of groundwater in agricultural watersheds | MODFLOW and MT3DMS | [19] |
| 2007 | Agriculture and groundwater nitrate contamination in the Seine basin. The STICS-MODCOU modelling chain | STICS-MODCOU-NEWSAM | [52] |
| 2007 | Factors Affecting the Spatial Pattern of Nitrate Contamination in Shallow Groundwater | Multivariate Tobit model | [53] |
| 2008 | Non-point pollution of groundwater from agricultural activities in Mediterranean Spain: the Balearic Islands case study | GIS-simulation model | [54] |
| 2008 | Modeling effects of nitrate from non-point sources on groundwater quality in an agricultural watershed in Prince Edward Island, Canada | 3-D two layer Numerical model (MT3DMS) | [55] |
| 2009 | Long-term nutrient leaching from a Swedish arable field with intensified crop production against a background of climate change | SOILN-DB | [56] |
| 2009 | Hydrochemical and stable isotopic assessment of nitrate contamination in an alluvial aquifer underneath a riverside agricultural field | Geochemical mass balance modeling | [57] |
| 2009 | Assessment of nitrate contamination of groundwater using lumped-parameter models | LPM | [58] |

**Table 1.** *Cont.*

| Year | Title | Model Used | References |
|------|-------|-----------|-----------|
| 2009 | Impact of fertilizer application and urban wastes on the quality of groundwater in the Cambrai Chalk aquifer, Northern France | Agri Flux, VS2DT-WHIUNSAT, MODFLOW | [59] |
| 2010 | Application of SWAT model to investigate nitrate leaching in Hamadan-Bahar Watershed, Iran | SWAT | [60] |
| 2010 | Nitrogen leaching in a typical agricultural extensively cropped catchment, China: experiments and modelling | LEACHMN | [61] |
| 2010 | Modeling Nitrate Leaching and Optimizing Water and Nitrogen Management under Irrigated Maize in Desert Oases in Northwestern China | WNMM | [62] |
| 2010 | Assessment of nitrogen contamination of groundwater in paddy and upland fields | PHREEQC and FEMWATER | [63] |
| 2011 | Spatial distribution pattern analysis of groundwater nitrate nitrogen pollution in Shandong intensive farming regions of China using neural network method | BPNN | [64] |
| 2011 | GIS-model based estimation of nitrogen leaching from croplands of China | DNDC Model–GIS | [65] |
| 2011 | Modelling the effect of forest cover in mitigating nitrate contamination of groundwater: A case study of the Sherwood Sandstone aquifer in the East Midlands, UK | MODFLOW–MT3DMS | [66] |
| 2011 | Long-term simulations of nitrate leaching from potato production systems in Prince Edward Island, Canada | LEACHM-MODFLOW | [67] |
| 2011 | Simulation of nitrate leaching under potato crops in a Mediterranean area. Influence of frost prevention irrigation on nitrogen transport | GLEAMS | [31] |
| 2011 | The effects of land use change and irrigation water resource on nitrate contamination in shallow groundwater at county scale | Semi variance models | [68] |
| 2012 | Assessment of the Intrinsic Vulnerability of Agricultural Land to Water and Nitrogen Losses via Deterministic Approach and Regression Analysis | GLEAMS and Multiple Regression | [25] |
| 2013 | Minimizing nitrate leaching while maintaining crop yields: insights by simulating net N mineralization | BOWAB | [69] |
| 2013 | Soil type, crop and irrigation technique affect nitrogen leaching to groundwater | ENVIRO-GRO (E-G) | [70] |
| 2013 | Nitrate leaching from a potato field using adaptive network-based fuzzy inference system | HYDRUS-2D and ANFIS | [27] |
| 2013 | Modeling of Nitrate Leaching from a Potato Field using HYDRUS-2D | HYDRUS-2D | [71] |
| 2013 | Modifying the LEACHM model for process-based prediction of nitrate leaching from cropped Andosols | LEACHM v/s LEACHM–RothC model | [72] |
| 2013 | Nitrate fluxes to groundwater under citrus orchards in a Mediterranean climate: Observations, calibrated models, simulations and agro-hydrological conclusions | Transient model | [73] |
| 2013 | Nitrate-Nitrogen Leaching and Modeling in Intensive Agriculture Farmland in China | LEACHM | [32] |
| 2014 | Calibration of DNDC model for nitrate leaching from an intensively cultivated region of Northern China | DNDC model | [28] |

**Table 1.** *Cont.*

| Year | Title | Model Used | References |
|------|-------|-----------|-----------|
| 2015 | Modelling nitrate pollution pressure using a multivariate statistical approach: the case of Kinshasa groundwater body, Democratic Republic of Congo | Multiple Linear Regression model | [74] |
| 2016 | Investigating nitrate dynamics in a fine-textured soil affected by feedlot effluents | HYDRUS-1D | [75] |
| 2017 | Simulating water and nitrogen loss from an irrigated paddy field under continuously flooded condition with Hydrus-1D model | HYDRUS 1D | [26] |
| 2019 | Modeling of Fertilizer Transport for Various Fertigation Scenarios under Drip Irrigation | HYDRUS-2D/3D | [76] |
| 2019 | Nitrate subsurface transport and losses in response to its initial distributions in sloped soils: An experimental and modelling study | HYDRUS-2D | [29] |
| 2019 | Groundwater Nitrate Contamination Integrated Modeling for Climate and Water Resources Scenarios: The Case of Lake Karla Over-Exploited Aquifer | MODFLOW | [77] |
| 2019 | Groundwater nitrate contamination in an area using urban wastewaters for agricultural irrigation under arid climate condition, southeast of Tehran, Iran | Simple Regression model | [78] |
| 2020 | Tracing nitrate sources in the groundwater of an intensive agricultural region | SIAR | [79] |
| 2021 | Quantifying nitrate leaching to groundwater from a corn-peanut rotation under a variety of irrigation and nutrient management practices in the Suwannee River Basin, Florida | SWAT | [30] |
| 2021 | A Spatially Distributed, Physically Based Modeling Approach for Estimating Agricultural Nitrate Leaching to Groundwater | FREEWAT | [80] |
| 2021 | Rotating maize reduces the risk and rate of nitrate leaching | APSIM | [81] |
| 2021 | Modelling effect of different irrigation methods on spring maize yield, water and nitrogen use efficiencies in the North China Plain. | WHCNS | [82] |
| 2021 | Modelling water consumption, N fates and maize yield under different water-saving management practices in China and Pakistan | WHCNS | [83] |
| 2022 | Spatiotemporal Modelling of Groundwater Flow and Nitrate Contamination in An Agriculture-Dominated Watershed. | MODFLOW-MT3DMS | [84] |
| 2022 | Modeling the water and nitrogen management practices in paddy fields with HYDRUS-1D | HYDRUS-1D | [85] |

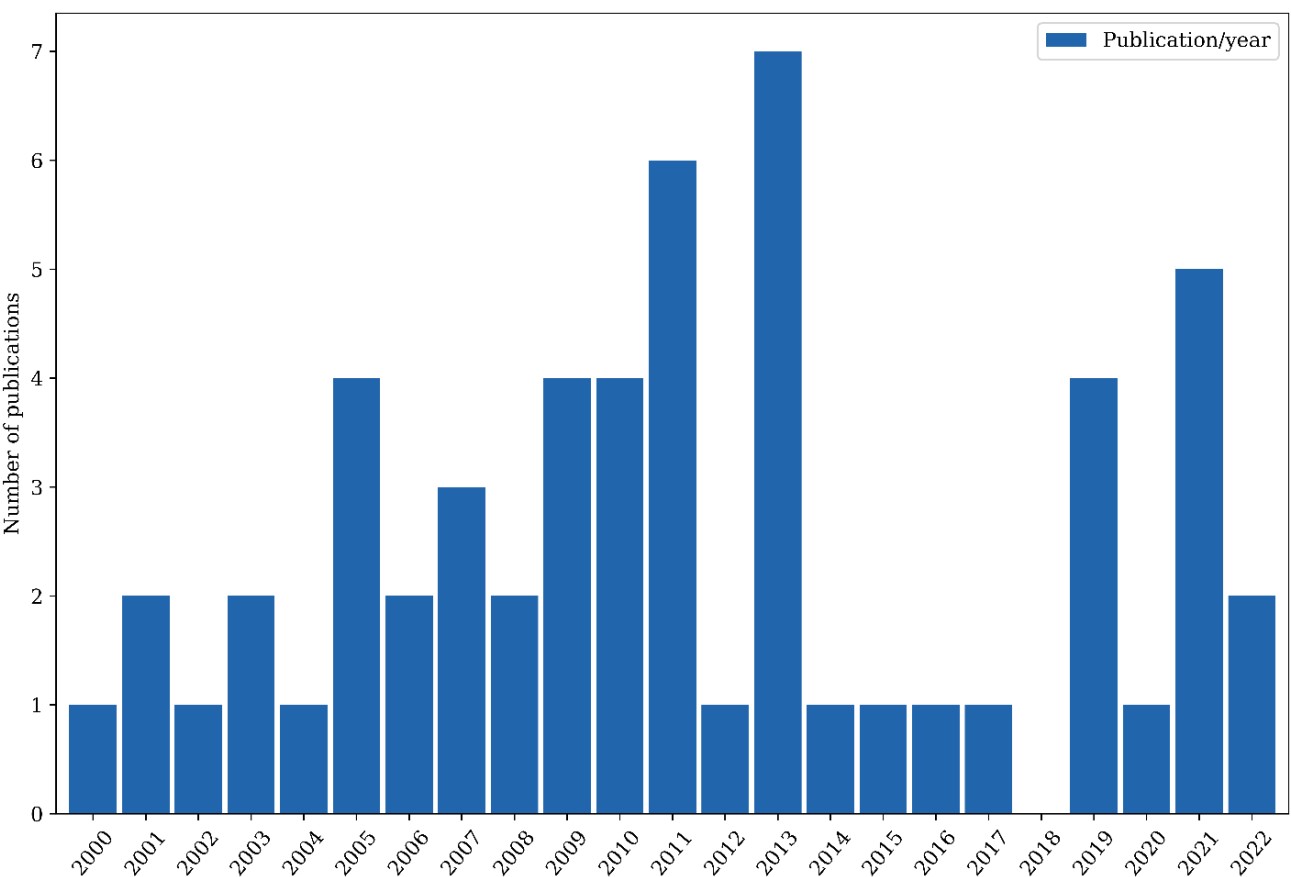

**Figure 4.** Temporal distribution of number of publications selected for SR per year.

*3.2. Model Complexities and Inputs*

We addressed model complexities in this study by assessing the input requirements for different models found in the selected publications. Based on model complexity and input variable requirements, we defined 15 input variables (Table 2).

"Soil properties" consist of information related to engineering, chemical, and physical properties of soil including pH, cation exchange capacity, particle size distribution, bulk density, and soil type. "Climate variables" consist of environmental factors like temperature and rainfall observations, relative humidity, solar radiation, evapotranspiration, and wind speed. "Management information and crop type" refers to planting date, fertilizer and irrigation applications, herbicides and organics amendments, and cultivar/variety, respectively. Input levels related to "land use", "soil hydraulic properties and geology", and "soil dispersivity" consist of what is growing on the land, the ease of water movement in the subsurface (hydraulic conductivity, porosity, hydraulic head, groundwater level, permeability, etc.), and the tendency of cohesive soils exposed to saturation by groundwater to separate into individual particles instead of forming small clumps or aggregates (i.e., soil diffusivity, dispersivity, fecal decomposition rate, litter decomposition rate, humus mineralization, etc.), respectively.

Some models reported in this study assimilate "groundwater data" referring to chemical constituents (Ca, Mg, Cl, $NO_3^-$, N, pH, EC, etc.) for model parametrization. "GIS layer" and "curve number" inputs were used by some models dealing with the parametrization of landscapes and slopes, soil water storage and surface runoff. "Percentage of vegetable, orchard, and barns fields" were used by some models to define the extent of land cover types.

**Table 2.** Input variables by different models studied as part of this SR and their definition.

| Variables | Descriptions |
|---|---|
| S | Soil Properties |
| CV | Climate Variables |
| MI | Management Information (i.e., planting date, fertilizer, irrigation, herbicides, organic amendment, etc.) |
| C | Crop Type |
| L | Land use |
| SH | Soil Hydraulic Properties (hydraulic conductivity, porosity, hydraulic head, groundwater level, permeability, etc.) |
| GD | Groundwater Data (Ca, Mg, Cl, $NO_3^-$, N, pH, EC, etc.) |
| GL | GIS Layer (Geological map, topographical map, shapefiles, soil map, other maps, etc.) |
| CN | Curve Number |
| VE | Percentage of vegetable fields |
| OR | Percentage of orchards |
| BA | Percentage of barns |
| SD | Soil dispersive and decomposition parameters (i.e., soil diffusivity, dispersivity, fecal decomposition rate, litter decomposition rate, humus mineralization, etc.) |
| F | Fertilizer |
| G | Geology |

Table 3 shows the summaries of models used in the selected publications, countries where the study was conducted, model type, and input variables. Model types were classified as integrated, process-based, numerical, GIS, geostatistical, mathematical, hydrogeochemical, and statistical. Results showed that integrated models are more complex due to model linkages and require more inputs than other models reported. Process-based models, though less complex than integrated models, showed similar attributes to numerical models reported in this study. GIS, geostatistical, mathematical, and statistical models reported in this study showed low input assimilation which does not always entail model accuracy. According to [86], model accuracy should allow us to choose the model that best approximates the physical observation and the confidence in the predictions of a given model. Specifying models based on complexities and input levels help researchers identify model suitability based on data availability and model application [87].

**Table 3.** Different input variables required by the models included in this study.

| Location | Model Used | Model Type | Input Variable (s) | References |
|---|---|---|---|---|
| Canada | LEACHM and statistical | Integrated | S + CV + MI + C + SD + SH | [41] |
| USA | NLEAP | Process | S + CV + MI + C | [42] |
| Egypt | GWS-3D | Process | SH + SD + G + F + CV | [43] |
| Spain | GLEAMS + PC-Arc/Cad GIS | Integrated | S + CV + MI + C + L | [44] |
| Morocco | Mathematical model | Numerical | S + SH + CV | [45] |
| Spain | GIS-GLEAMS | Integrated | S + CV + MI + C + L | [24] |

**Table 3.** *Cont.*

| Location | Model Used | Model Type | Input Variable (s) | References |
|---|---|---|---|---|
| Japan | GIS | GIS | GD + GL | [46] |
| USA | NRCS CN model | Mathematical | S + CN | [47] |
| Netherland | ANIMO, and STONE | Integrated | L + MI + S+CV | [48] |
| Turkey | NLEAP | Process | S + CV + MI + C | [18] |
| USA | LEACHMN | Process | S + CV + MI + C | [49] |
| USA | HYDRUS-2D | Process | SH + S+CV | [50] |
| Germany | Numerical model | Numerical | S + SH + CV | [51] |
| USA | MODFLOW and MT3DMS | Integrated | L + GL + S + CV + SH | [19] |
| France | STICS-MODCOU-NEWSAM modelling chain | Integrated | S + CV + MI + C+GD + L+GL | [52] |
| Korea | Multivariate Tobit model | Statistical | VE + OR + BA + GD | [53] |
| Spain | GIS-simulation model | GIS | GD + GL | [54] |
| Canada | 3-D two layer Numerical model (MT3DMS) | Numerical | L + GL + S + CV + SH | [55] |
| Sweden | SOILN-DB | Process | CV + MI + SD | [56] |
| Korea | Geochemical mass balance modeling | Hydro-geochemical | GD + GL | [57] |
| Palestine | LPM | Process | GL + SH | [58] |
| France | Agri Flux, VS2DT-WHIUNSAT, MODFLOW | Integrated | L + GL + S + CV + SH | [59] |
| Iran | SWAT | Process | S + C + CV + MI + GL + SH | [60] |
| China | LEACHMN | Process | S + CV + MI + C + SD + SH | [61] |
| China | WNMM | Process | SH + S + CV + MI + SD | [62] |
| Taiwan | PHREEQC and FEMWATER | Numerical | S + SH + SD | [63] |
| China | BPNN | Statistical | S + GD + F | [64] |
| China | DNDC-GIS | Integrated | GL + S + C + MI + CV | [65] |
| UK | MODFLOW-MT3DMS | Integrated | L + GL + S + CV + SH | [66] |
| Canada | LEACHM-MODFLOW | Integrated | S + CV + MI + C + SD + SH + GL + L | [67] |
| Spain | GLEAMS | Process | S + CV + MI + C | [31] |
| China | Semi-variance models | Statistical | GD + GL | [68] |
| USA, Italy, Greece | GLEAMS and Multiple regression | Integrated | S + CV + MI + C + GD + L + GL + CN + SH | [25] |
| Germany | BOWAB | Process | S + C + CV + MI | [69] |
| USA | ENVIRO-GRO (E-G) | Process | S + C + CV + MI | [70] |
| Iran | HYDRUS-2D and ANFIS | Integrated | SH + S + CV | [27] |
| Iran | HYDRUS-2D | Process | SH + S + CV | [71] |
| Japan | LEACHM v/s LEACHM-RothC model | Integrated | S + CV + MI + C + SD + SH | [72] |
| Israel | Transient model | Numerical | CV + S + SH + GD | [73] |
| China | LEACHM | Process | S + CV + MI + C | [32] |
| China | DNDC model | Process | GL + S + C + MI + CV | [28] |

**Table 3.** *Cont.*

| Location | Model Used | Model Type | Input Variable (s) | References |
|---|---|---|---|---|
| Congo | Multiple Linear Regression model | Statistical | SH + G + GL + L + GD | [74] |
| Argentina | HYDRUS-1D | Process | SH + S + CV | [75] |
| China | HYDRUS-1D | Process | SH + S + CV | [26] |
| Egypt | HYDRUS-2D/3D | Process | SH + S + CV | [76] |
| Greece | MODFLOW | Numerical | L + GL + S + CV + SH | [29] |
| Iran | Simple regression model | Geostatistical | GD + GL | [77] |
| China | HYDRUS 2D | Process | SH + S + CV | [78] |
| China | SIAR | Statistical | GD | [79] |
| USA | SWAT | Process | CV + MI + C + GL + G + SH + L | [30] |
| Italy | FREEWAT | Integrated | L + MI + S + CV | [80] |
| USA | APSIM | Process | S + C + CV + MI | [81] |
| China | WHCNS | Process | S + C + CV + MI | [82] |
| China and Pakistan | WHCNS | Process | S + C + CV + MI | [83] |
| USA | MODFLOW and MT3DMS | Integrated | L + GL + S + CV + SH | [84] |
| China | HYDRUS-1D | Process | SH + S + CV | [85] |

*3.3. Spatio-Temporal Model Distribution*

The temporal and spatial distribution of different models considered in this study is presented in Figures 5 and 6, respectively. It was observed that computer-based models have gained popularity among the research community to simulate physical conditions in the area of groundwater nitrate contamination. For example, from 2001 to 2010 process-based models were found to dominate (Figure 5), implying that most researchers in the early 21st century used process-based models. However, results for the recent decade showed that model selection and relevance are variable with marginal differences in the order process-based > integrated > statistical = numerical. It is also important to note that the year 2013 had the highest number of model use, with 2 integrated, 4 process-based, and 1 numerical model reported in the selected publications. However, based on yearly model distribution, integrated models were used 3 times in 2011, twice in 2007 and 2013, and once in 2000, 2003, 2005, 2012, 2021, and 2022. Process-based models were used 4 times in 2013 and 2021, 3 times in 2010, twice in 2001, 2005, 2009, and 2019, and once in 2002, 2006, 2011, 2014, 2016, 2017, and 2022.

GIS, mathematical, and hydrogeological models have only been used once in 2004 and 2008, 2003 and 2005, and 2009, respectively. We observed that statistical model usage was more consistent in the recent decade than in the previous decade, with these types of models used twice in 2011, and once each in 2007, 2015, 2019, and 2020. Similar trend was also observed for the numerical models in this study. In general, process-based models were the most used, followed by integrated models, then statistical and numerical models.

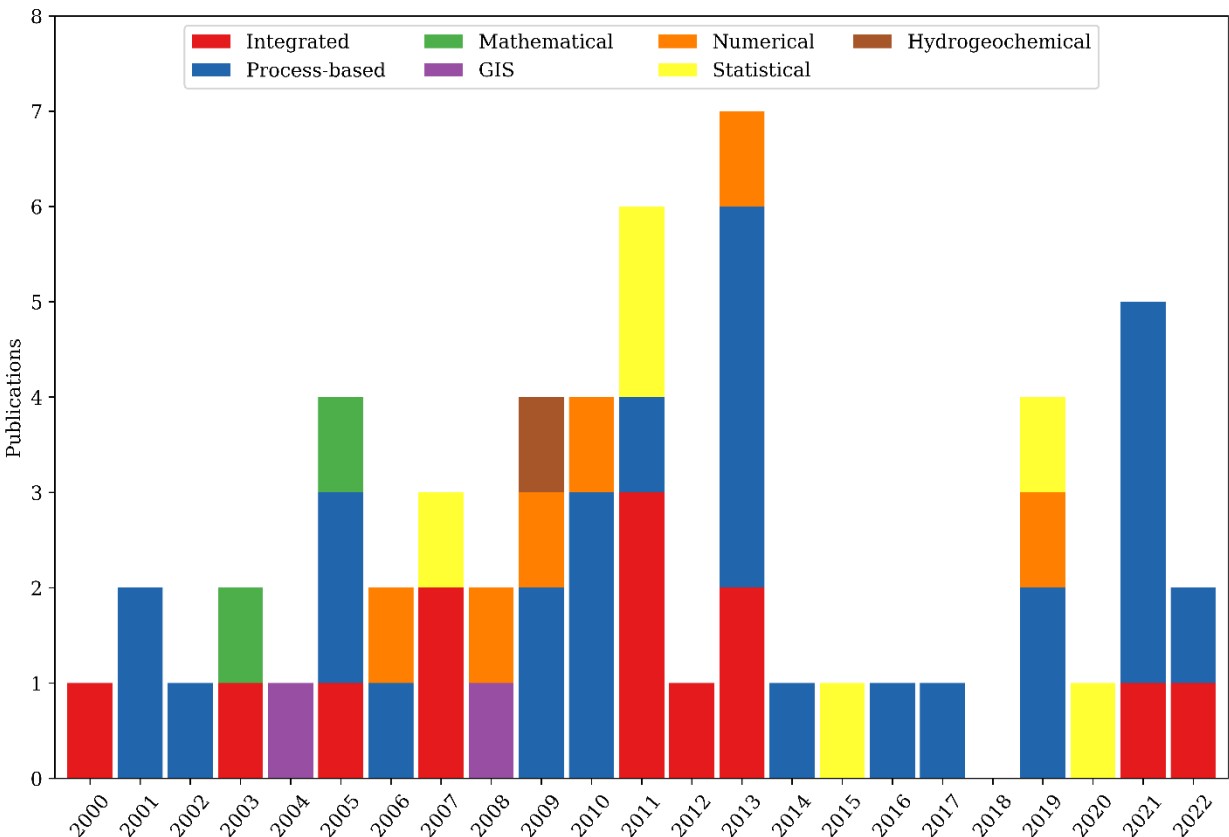

**Figure 5.** Number of publications that used different model types from the years 2000 to 2022.

Results of spatial distribution of models around the world showed that China dominantes in the use of process-based models (Figure 6) more than any other country reported in this study, followed by North America (the US and Canada). Process-based models and integrated models combined were mostly used in North America, (Figure 6). The dominance of process-based and integrated models in China, United States, and Canada may suggest the progression of modeling nitrate contamination of groundwater in these regions. In addition, this could also indicate the availability of resources for field data collection needed for calibrating and validating these complex models. Similar pattern of process-based and integrated model usage, although lower in number as compared to North America and China was observed in the European Union (including France, Spain, UK, Greece, Italy, Germany, Turkey, Netherlands, and Sweden). Distribution of statistical model use showed dominance in China, followed by South Korea, Iran, and Central Africa. This may suggest a preference or relative confidence in the use of these model types in the region. It is also interesting to note that relatively simple statistical models dominated Central Africa while process-based and numerical models dominated North Africa (Egypt and Morocco) and the Middle East (Iran). The regional trend of selection of different model types could be due to several reasons including but not limited to: (i) availability of research funding to collect large and intense input datasets, (ii) experience and previous knowledge of a specific model, (iii) research focus on advancing nitrate contamination research, (iv) the physical origin of the development of a particular model/models. For instance, a study [88] observed that researchers have an implicit advantage to use models developed in their regions. This is true for DSSAT model, whichwas originally developed in the United States [89] and has been serving crop modelers in the United States for decades, even though other crop models like STICS [90], WOFOST [91], CropSyst [92], and CROPWAT [93] exist.

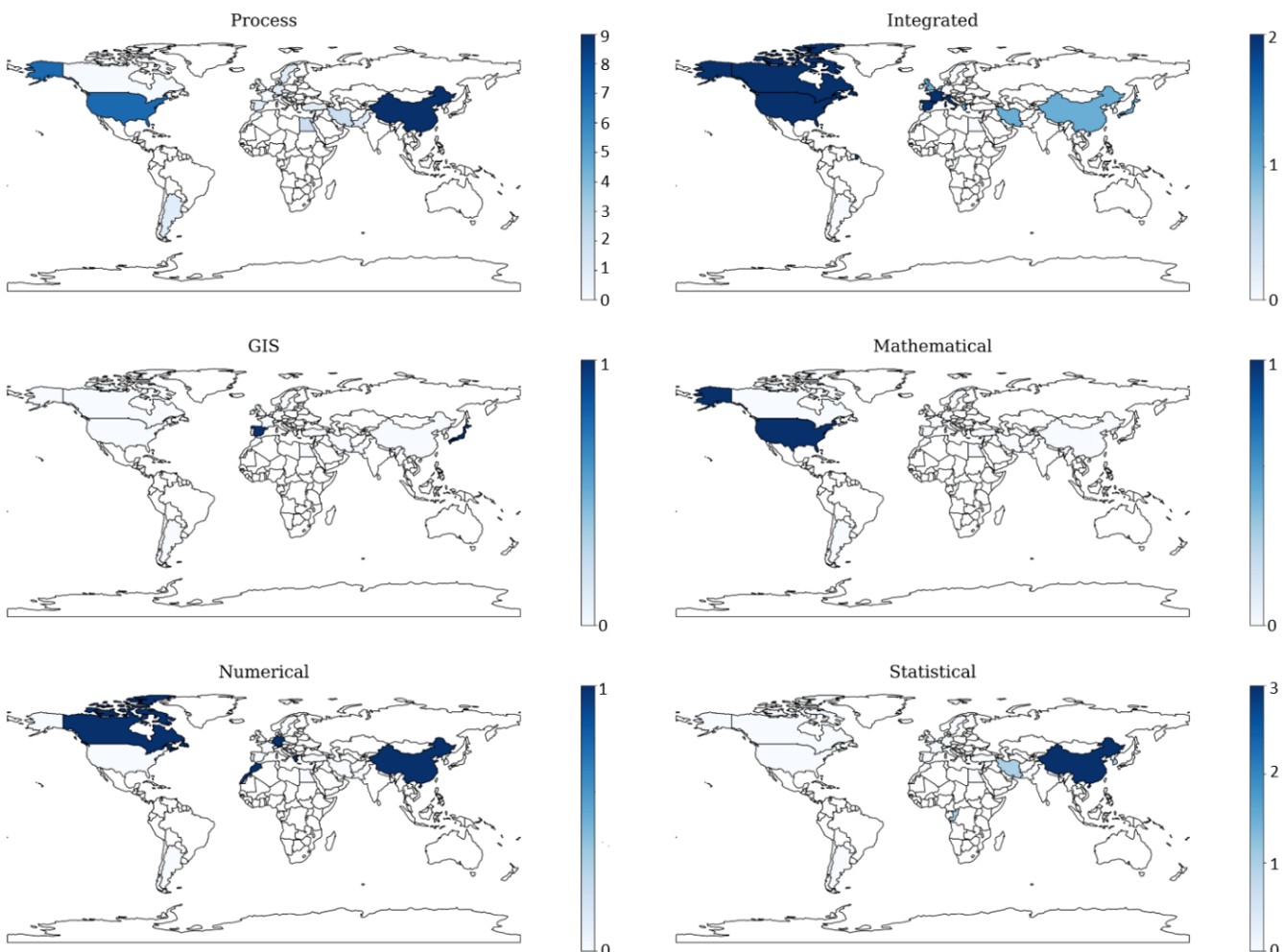

**Figure 6.** Spatial distribution of different model types (except Hydrogeological) used all over the world.

For countries and regions reported previously, half of the studies used process-based models. In the following section, we focus on which process-based models are predominantly used in the selected publications.

### 3.4. Model Selection and Evaluation Metrics

Evaluation metrics include parameters used to assess a model's predictive accuracy. All the evaluation metrics used in the selected publications and the number of times they were used are listed in Table 4. We observed that Root Mean Square Error (RMSE) is the most commonly used evaluation metric in the selected publications, followed by correlation coefficient (r) and coefficient of determination ($R^2$). This observation is consistent with a study [94] that found RMSE to be a widely used evaluation metric. It is important to note that not all selected studies reported different kinds of evaluation metrics used, which posed a problem in generalizing or creating a subset of statistical evaluation metrics. Nonetheless, many selected studies confirmed the widespread use of RMSE as compared to other reported metrics to identify the accuracy of model predictions.

**Table 4.** Evaluation metrics used in the models.

| Key | Evaluation Metrics | Number of Times Used |
|---|---|---|
| RMSE | Root mean square error | 19 |
| r | Correlation coefficient | 14 |
| $R^2$ | Coefficient of Determination | 9 |
| MAE | Mean absolute error | 5 |
| NSE | Nash-Sutcliffe modeling efficiency | 4 |
| E | Coefficient of efficiency | 3 |
| d | Index of agreement | 3 |
| RMS | Root mean squared | 1 |
| MAPE | Mean absolute percentage error | 1 |
| MED | Mean error difference | 1 |
| CRM | Coefficient of residual mass | 1 |
| ME | Mean error | 1 |
| RE | Relative error | 1 |
| PBIAS | Percent bias | 1 |

Process-based model HYDRUS was the most used model for simulating nitrate contamination of groundwater due to agricultural activities followed by LEACHM, NLEAP, SWAT, and WHCNS (Figure 7). This could indicate that HYDRUS is easy to use and parametrize and is accurate in reproducing observed information. Availability of documentation and help could be other factors contributing to the popularity of HYDRUS along with its capability to model several different processes as reported by several studies [95,96].

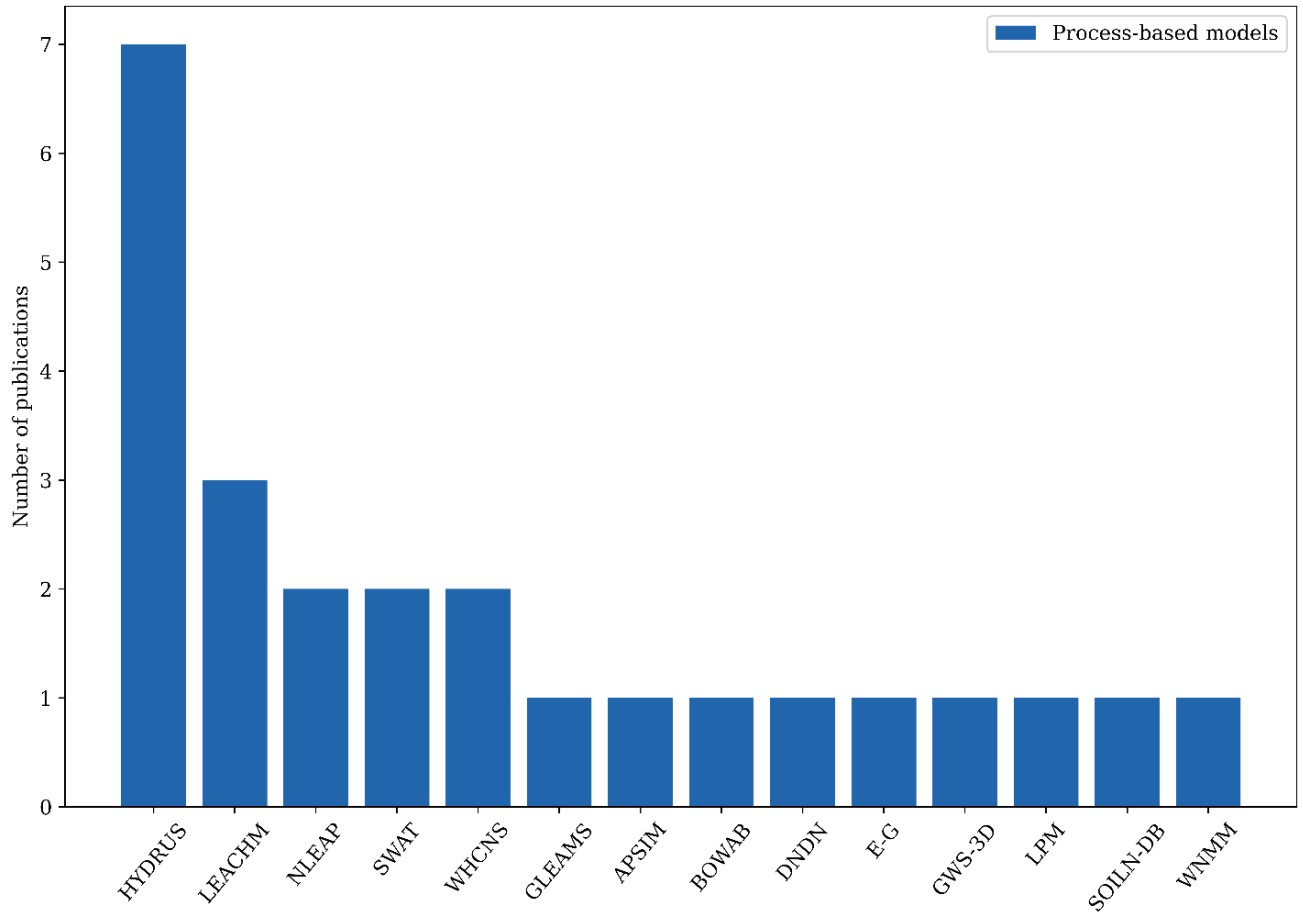

**Figure 7.** Number of publications included in this SR that used different process-based models.

*3.5. Challenges of Modeling Groundwater Nitrate Contamination*

We observed that the main challenges in modeling groundwater nitrate contamination due to agricultural activities are the decision on which model to use, in-season observation for calibration and validation of different models, availability of input data at field scale including climate information and soil data, and model evaluation metrics. This list of challenges is not exhaustive as model needs vary. Different models have different levels of input needs, with integrated and process-based models requiring large number of inputs in comparison to a simple statistical model. Although, statistical models in the form of machine learning algorithms have proven to be more data-intensive than process-based and integrated models combined [97], using these kinds of models may pose a huge challenge in countries where data are not readily available.

In addition, choosing model evaluation metrics revealed another challenge in the literature studied. We observed a lack of consensus in the choice of evaluation metrics though most studies preferred to use RMSE. We believe that depending on the scope of the study this SR can help guide and reduce the burden of model selection modeling groundwater nitrate contamination due to agricultural activities.

## 4. Discussion and Conclusions

By conducting a systematic review on the modeling of groundwater nitrate contamination due to agricultural activities, we provided a comprehensive overview of various kinds of models used in this domain worldwide, as well as input requirements and evaluation metrics used by different models. This study showed that model usage differs in scale, model type, and preference of the researcher. Moreover, it indicates that some models are used more than others worldwide, though we cannot conclude which is the best model in this domain.

We found that China, North America and the European Union focused more on process-based and integrated models. We also found that advanced statistical and machine learning models are thriving more in China than any other country reported in this study. Model complexity and input requirements were used to stratify different kinds of models, with integrated model being most complex and input-intensive, while simple statistical models being less input-intensive. The use of process-based and integrated models found more applications in the developed countries of China, North America, and Europe, while simple statistical model found applications in developing countries. Spatio-temporal model distribution results revealed that studies on groundwater nitrate contamination due to agricultural activities have increased in the recent decade (2011–2022) than the decade before it. We observed that researchers have implicit advantage in using models developed in their regions. In addition, our study found that process-based models have been most widely used in the last two decades, followed by integrated models. The results indicate that HYDRUS models are the most used for simulating nitrate contamination of groundwater due to agricultural activities, followed by the LEACHM model. Evaluation and assessment metrics results concluded that root mean square error (RMSE) is the most preferred metric followed by correlation coefficient (r) and coefficient of determination ($R^2$), in that order.

We recognize the shortcomings of this SR based on several key decisions that we had to make. First, we chose to include only those peer-reviewed articles that were written in English and had full texts available online. Therefore, this SR does not include any modeling of nitrate groundwater contamination due to agricultural activities that was presented in conference proceedings or textbooks. We believe that only including articles written in English also creates a bias towards English speaking/reporting regions while underrepresenting others. Due to the wide variety of model types, input data available and used, data sources, methodologies, and evaluation metrics used, we faced several challenges in identifying an appropriate classification system to represent and summarize the main findings. The steps used and criteria developed were revised several times though we recognize that it's likely that this SR might not be fully representative.

Despite these limitations, we believe that this SR summarized the state of the art in use of models for studying nitrate contamination of groundwater due to agricultural activities. The authors conclude that this topic is complex and multi-dimensional, and this study provides direction and a toolset for making better informed decisions regarding choosing models in this domain.

**Author Contributions:** Conceptualization, M.R., R.S., I.O. and T.W.; methodology, M.R., R.S., I.O. and T.W.; Writing—original draft preparation, M.R.; Writing—review and editing, M.R., R.S., I.O. and V.S.; Supervision, V.S.; Project administration, V.S. All authors have read and agreed to the published version of the manuscript.

**Funding:** This research received no external funding.

**Institutional Review Board Statement:** Not applicable.

**Informed Consent Statement:** Not applicable.

**Data Availability Statement:** Not applicable. All data generated or analyzed during the study are included in this review article.

**Conflicts of Interest:** The authors declare no conflict of interest.

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
