# Peer review of "Modeling of Groundwater Nitrate Contamination Due to Agricultural Activities—A Systematic Review"

_water, doi:10.3390/w14244008_

Round 1

Reviewer 1 Report

This study reported the modeling of groundwater nitrate contamination. The manuscript was well written, and the methodology was well documented. However, the study was too simple, and lacked scientific significance for the relevant topic. I am skeptical that this study has provided or clarified any mechanistic understanding. More in-depth discussion is needed. Different models were not compared in regards to model complexity, model performance, model selection in specific circumstances, etc. Given the title of ‘modeling of groundwater nitrate contamination due to agricultural activities’, the effect of agricultural activities on groundwater nitrate contamination in relation to modeling was not discussed. The sample size of 49 is hardly representative especially when comparing modeling in different countries around the world. Intensive revisions are needed before this manuscript can be published on Water.

Reviewer 2 Report

This problem is relevant for journal scope. 

I could find some typing errors. The topic of the article is up to date, the introduction and literature survey is easy detailed. The concept and aim not are clearly defined. 

However, the literature review has a limited presentation, is based on  articles and does not offer the basis to formulate the objectives of research.

The objectives of research and the study novelty, in comparison with international relevant literature are missing.

The presentation and discussion of the result is clear and very detailed. Lack of conclusions in this paper. The conclusions are not well extracted from the results and discussion. Despite a good analysis of the literature review, there is no definite information gathered in conclusions.

Remarks and suggestions:

The introduction needs to be improved

Scientific objectives should be defined

Present keywords in alphabetical mode

Literature review is not a keyword

Reviewer 3 Report

Dear Authors

I have now completed the review of the manuscript titled “Modeling of Groundwater Nitrate Contamination Due to Agricultural Activities – A Systematic Review”. This Systematic Review (SR) seeks to provide a comprehensive overview of different models used to estimate nitrate contamination of groundwater due to agricultural activities. The authors described different types of models available in the field of modeling groundwater nitrate contamination, the models used, the requirement of input parameters for models, and the evaluation metrics used. Out of all the models reviewed, stand-alone, process-based models are predominantly used for modeling nitrate contamination, followed by integrated models with HYDRUS model followed by LEACHM being the two most commonly used process-based models worldwide. The topic is quite interesting and relevant. I have a few comments to improve the quality and clarity of the manuscript.

1.     Line 27-29: There is a need… achieve this, this statement should have additional references such as [1-2].

2.     A fundamental improvement in this investigation is to add the works up to 2022.

  1.  Figure 3. Authors can add [3-4].
  2. Line 365-367: However, with advances in statistical modelling, machine learning processes have emerged as powerful predictive models… data intensive. I am quite afraid of this statement. The data is growing at a great pace, and using excel is fine but embracing AI/ML/statistical models is highly required along with computer vision, please see and add 2 and 5.
  3. Table 3 should be supported by references used in the investigation.
  4. Line 328: Authors can also add Cropwat.
  5. The paper ends with the discussion, a new section Conclusion is required which talks about the findings, limitations and major takeaways from the investigation

Overall the article is discussing a very important aspect, however, it requires improvement and clarity based on the above points.

References

1. United Nations, The United Nations World Water Development Report 2022: Groundwater: Making the invisible visible. UNESCO, Paris.

2. Crop Water Requirements with Changing Climate in an Arid Region of Saudi Arabia

3. Anthropogenic nitrate in groundwater and its health risks in the view of background concentration in a semi arid area of Rajasthan, India

4. Hydrochemical characteristics and quality assessment of shallow groundwater under intensive agriculture practices in arid region, Qena, Egypt

5. Deep Learning Based Modeling of Groundwater Storage Change, cmc.

Round 2

Reviewer 3 Report

In the manuscript titled "Modeling of Groundwater Nitrate Contamination Due to Agricultural Activities – A Systematic Review"  the authors have incorporated the suggestions.